# AI Should Sense Better, Not Just Scale Bigger: Adaptive Sensing as a Paradigm Shift

Eunsu Baek[*1], Keondo Park[1], JeongGil Ko[2], Min-hwan Oh[1], Taesik Gong[3], and Hyung-Sin Kim[†1]

[1]Graduate School of Data Science, Seoul National University, Seoul, South Korea
[2]School of Integrated Technology, Yonsei University, Seoul, South Korea
[3]Department of Computer Science and Engineering, UNIST, Ulsan, South Korea
[1]{beshu9407, gundo0102, minoh, hyungkim}@snu.ac.kr
[2]jeonggil.ko@yonsei.ac.kr
[3]taesik.gong@unist.ac.kr

## Abstract

Current AI advances largely rely on scaling neural models and expanding training datasets to achieve generalization and robustness. Despite notable successes, this paradigm incurs significant environmental, economic, and ethical costs, limiting sustainability and equitable access. Inspired by biological sensory systems, where adaptation occurs dynamically at the input (e.g., adjusting pupil size, refocusing vision)—we advocate for *adaptive sensing* as a necessary and foundational shift. Adaptive sensing proactively modulates sensor parameters (e.g., exposure, sensitivity, multimodal configurations) at the input level, significantly mitigating covariate shifts and improving efficiency. Empirical evidence from recent studies demonstrates that adaptive sensing enables small models (e.g., EfficientNet-B0) to surpass substantially larger models (e.g., OpenCLIP-H) trained with significantly more data and compute. We (i) outline a roadmap for broadly integrating adaptive sensing into real-world applications spanning humanoid, healthcare, autonomous systems, agriculture, and environmental monitoring, (ii) critically assess technical and ethical integration challenges, and (iii) propose targeted research directions, such as standardized benchmarks, real-time adaptive algorithms, multimodal integration, and privacy-preserving methods. Collectively, these efforts aim to transition the AI community toward sustainable, robust, and equitable artificial intelligence systems.

## 1 Introduction

Early neural-network research drew inspiration from cortical neurons, implicitly framing intelligence as a predominantly brain-centered phenomenon [49, 67]. Correspondingly, recent advancements in artificial intelligence (AI) have predominantly relied on scaling model size and expanding training datasets [78, 2, 82, 10]. Although effective, this *model-centric paradigm* introduces significant and unsustainable challenges, such as massive computational demands that exacerbate environmental degradation [14, 59], socioeconomic inequities due to resource concentration in a few well-funded institutions [12, 13], and persistent failures in generalizing under real-world domain shifts [29, 39].

In contrast, biological cognition is fundamentally embodied: the brain operates within an integrated system comprising sensory organs (e.g., vision, hearing, and touch) and musculature, which forms critical perception-action loops essential for survival and evolutionary success. For robust perception, human sensory systems dynamically adapt at the sensor level both before and during cortical processing. For instance, humans correct optical blur by wearing corrective lenses rather than retraining their neural circuits with thousands of out-of-focus images. Similarly, when facing low contrast or glare, humans instinctively squint, rapidly narrowing the pupil to sharpen depth-of-field and suppress stray light. Thus, intelligence emerges from the coordinated evolution and dynamic interplay

---

[*]edw2n.github.io. [†]Corresponding author

between sensory organs, neural pathways, and adaptive behaviors, not solely from neural complexity alone [37, 74]. This biological principle highlights that effective perception and generalization in AI depend **not only on neural capacity but also on dynamic, real-time sensor adaptation** throughout the entire perception pipeline.

However, current AI methodologies still primarily address distribution shift by *scaling* neural architectures and datasets [55, 17, 2, 64], leaving the sensing interface unchanged. A few existing sensor-aware strategies, such as active perception, which repositions sensors or robots to optimize viewpoints [11, 73], and sensor-fusion budgeting, which schedules when and what modalities to activate [79], operate at the system or motion-planning level. Post-hoc physics-based [3, 60] and physics-informed (e.g., PINNs, DSE) [65, 46] sensor simulations cannot recover the coupled environment–sensor dynamics once analog signals are digitized, as the measurements have already been shaped by sensor configurations, leading to irreversible information loss. They neglect how raw analog signals are digitized in the first place to best serve the downstream model. Historically, adaptive optics (e.g., radar CFAR [21, 66], astronomical seeing correction [7]) were designed to enhance human perception or measurement fidelity, but were grounded in physical or human-centric criteria rather than how sensors adapt to model's representation space.

Only recently have benchmarks begun isolating the impact of intrinsic sensor settings, such as exposure and gain, on recognition accuracy [9, 8]. Inspired by these insights, the concept of **test-time input adaptation** has emerged: dynamically optimizing sensor parameters frame-by-frame to deliver model-friendly data. A first prototype, Lens [8], significantly outperforms traditional (human-oriented) auto-exposure methods, maintaining high classification accuracy despite a $50\times$ reduction in model size. Preliminary evidence suggests even greater potential: with ideal sensor adaptation, a lightweight 5M-parameter EfficientNet-B0 [47] can surpass the 632M-parameter OpenCLIP-H [33] trained on $160\times$ more training data [9]. These early results highlight both the promise of adaptive sensing and the urgency for systematic exploration.

While initial empirical demonstrations come from image-classification tasks, adaptive sensing is broadly applicable across diverse domains. By shaping incoming photons, acoustic waves, or tactile forces *at the sensor*, it boosts efficiency and robustness in fields such as medical diagnostics [61, 5], autonomous driving [69], surveillance [86], and environmental monitoring [4, 48]. This capability is becoming critical as AI increasingly transitions from artificial, controlled simulations to **physical, embodied deployments** [57]. In real-world robots and wearable devices, identical models may receive data from lenses, CCDs, MEMS microphones, or tactile skins, each differing across vendor, firmware, and production batch. These subtle differences introduce domain shifts, either explicit or latent, that are difficult to address through model advancement alone.

Recent embodied-AI benchmarks in locomotion [71, 43], household manipulation [75], and interactive task execution [91] exemplify an urgent reality, where the future AI must sense, reason, and interact from and with the real world within strict real-time and on-device constraints. Physical AI platforms from micro-drones to neural prostheses face inherent limits in size, energy, and thermal dissipation, suggesting limits to the traditional "bigger model, bigger dataset" paradigm. Adaptive sensing provides a **complementary solution**: by reshaping inputs directly at the source *(i.e., at the hardware level)*, it delivers model-friendly signals that enable smaller networks to thrive. Just as the human visual system couples rapid eye movements to visual cognition, next-generation AI systems must integrate closed-loop, real-time sensor control and preprocessing into their inference pipeline.

**Our Position.** Inspired by the principle in biological sensory systems, we argue that **AI research must transition away from an exclusively model-centric paradigm toward prioritizing adaptive, input-level optimization as a first-class concern.** Adaptive sensing is more than an incremental advance. Rather, it represents a critical, paradigm-level shift toward sustainable, equitable, and robust AI. Sections 2-7 articulate key research directions, evaluation frameworks, and interdisciplinary opportunities essential to establish adaptive sensing as a foundational pillar of future AI.

## 2 Current State of AI: Limitations of the Model-Centric Paradigm

The dominant approach in modern AI emphasizes scaling models and expanding datasets to improve robustness and generalization [14, 13]. Although this model-centric paradigm has yielded notable performance gains, it faces fundamental limitations that increasingly threaten its long-term viability:

- **Environmental and Computational Cost:** Modern AI models, particularly large-scale models such as GPT-3/4 [14, 2], and advanced multimodal models [78, 81, 82], require massive compu-

tational resources for training [18]. Training GPT-3 alone requires approximately 1.287 GWh of electricity, emitting roughly 552 tons of $CO_2$, comparable to driving a passenger vehicle for more than one million kilometers [59, 76]. The ongoing trend toward even larger models [38, 30] exponentially escalates these environmental costs, posing significant sustainability challenges.

- **Accessibility and Inequity:** The heavy computational requirements of training and deploying state-of-the-art models (e.g., GPT variants, CLIP, and advanced multimodal transformers) restrict access primarily to organizations with substantial financial and computational resources [13]. This centralization widens the global digital divide by disproportionately benefiting wealthy institutions and regions while restricting innovation and participation from smaller-scale researchers, startups, and economically disadvantaged groups. Consequently, the potential diversity of ideas and equitable distribution of AI benefits remain severely constrained.

- **Real-World Generalization Failures:** AI models trained on massive yet static datasets frequently fail to generalize effectively when encountering novel or dynamically changing real-world conditions. Domain shifts–variations in environmental, sensor-specific, and task-specific contexts–are notoriously challenging for traditional data-centric approaches [63]. Existing robustness benchmarks inadequately capture the true complexity of real-world conditions [9]. As a result, models trained under these benchmarks often exhibit significant performance drops when deployed in realistic settings, limiting their reliability in critical applications such as autonomous driving [69], medical diagnostics [61, 5] and environmental monitoring [4, 48].

- **Economic Sustainability and Scalability:** The continuous increase in model complexity and training data volumes imposes a substantial financial burden [76, 16]. The costs associated with advanced computational infrastructure, specialized hardware, and extensive data acquisition quickly become prohibitive. This economic barrier hampers the scalability and widespread adoption of advanced AI technologies, especially in resource-constrained settings or smaller-scale enterprises.

- **Ethical and Societal Concerns:** Large-scale models inherently risk amplifying biases present in extensive, complex training datasets [27, 40]. As data volumes grow, auditing, identifying, and mitigating embedded biases becomes increasingly difficult [70]. Without rigorous oversight, these biases can propagate societal harm, perpetuating stereotypes, inequities, and systemic prejudices [12]. Addressing these ethical implications through a model-centric lens alone remains insufficient, reinforcing the need for alternative paradigms.

Collectively, these limitations emphasize the urgency of moving beyond a solely model-centric paradigm toward strategies that integrate smarter, adaptive sensing and context-aware optimization.

## 3   Adaptive Sensing as a Necessary Paradigm Shift

Overcoming the limitations of the model-centric paradigm requires treating sensor-level adaptability as a first-class design principle. While biological sensors embed rapid, energy-proportional adaptation in hardware, modulating gain, bandwidth, and spatial resolution before neural inference begins, modern artificial sensors, such as cameras, microphones, and haptic arrays, **remain predominantly static** (Table 1). Bridging this gap through adaptive sensing at the input level would enable smaller, faster, and fairer AI models, providing a compelling path toward sustainable and equitable artificial intelligence.

Table 1: Comparison between adaptive human sensors and static artificial sensors.

| Modality | Human Sensor (Adaptive) | Artificial Sensor (Static) |
|---|---|---|
| **Vision** | Pupil diameter 2–8 mm ($\sim16\times$ light gain) in < 200 ms. Dark adaptation restores sensitivity; ciliary muscles refocus from 10 cm to infinity. Saccades (3–5 ms) redirect fovea before cortex [47, 87]. | Fixed or two-step aperture; ISO/shutter in coarse steps [26]. Autofocus 50–300 ms; quantum efficiency and CFA static. Saturation corrected only post-capture. |
| **Hearing** | Active gain control (> 120 dB dynamic range) via outer-hair cells [42]. Real-time efferent feedback loop protects hearing from damage [52]. 3-D localization from binaural delay down to 10 μs. | 60-90 dB dynamic range (MEMS mic) with fixed AGC [72]. Limited impulse protection. 3D localization requires multi-element arrays and DSP. |
| **Touch** | Sensitivity of 0.3-500+ kPa via mosaic of mechanoreceptors. Spatial acuity up to 0.5mm; temporal resolution $\sim$ 1kHz [83, 36]. Soft, compliant skin spreads out pressure; nociceptors trigger pain reflexes. | Sensitivity of 1-100 kPa via capacitive or piezoresistive arrays. Taxel pitch 1mm; sampling 100-500Hz [25]. Mechanically rigid surface with no pain feedback [34]. |

### 3.1   Early Evidence from Vision Tasks

Recent work on ImageNet-ES [9] and ImageNet-ES-Diverse [8] evaluates real-world covariate shifts induced by controlled sensor variation. Lens [8] (Figure 1) is the first model-friendly, test-time input adaptation framework to mitigate covariate shifts for image classification. It operates in a post-hoc, adaptive, and camera-agnostic sensor control manner, dynamically responding to scene characteristics based on VisiT scores to provide optimal image quality for neural networks. The key empirical

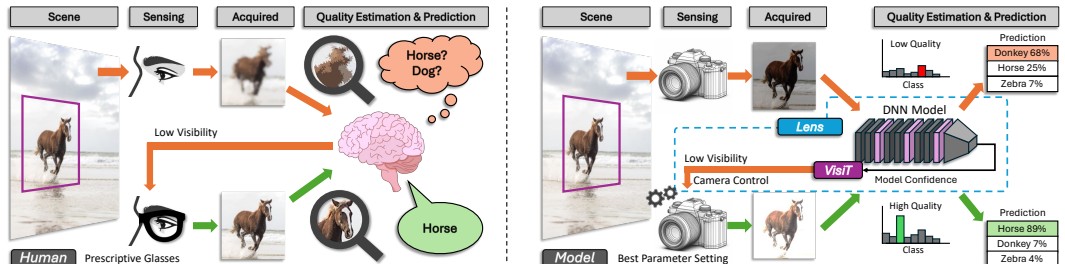

Figure 1: A representative adaptive sensing framework (Lens [8]).

Table 2: Broad real-world applicability of adaptive sensing.

| Domain | Role of Adaptive Sensing |
|---|---|
| **Humanoids** | Adaptive multimodal sensor adjustments—such as dynamically modulating visual, auditory, and tactile sensors—improve real-time balance, manipulation, and interaction quality in complex, unstructured environments. |
| **Healthcare** | Dynamic adjustment of medical imaging parameters optimizes image quality based on patient-specific characteristics and contexts (e.g., dynamically adjusting MRI sequences to enhance diagnostic accuracy). |
| **Self-driving** | Real-time optimization of camera and lidar parameters allows rapid adaptation to changing lighting and weather (e.g., fog, rain, or sudden brightness changes), enhancing perception robustness and road safety. |
| **Agriculture** | Adaptive drone sensing systems dynamically adjust settings to capture high-quality data under varying conditions—such as crop type, growth stage, and environmental stress—for precise crop health monitoring. |
| **Environment** | Adaptive sensing dynamically tunes sensor settings to capture accurate air and water quality data under diverse environmental conditions, improving monitoring accuracy and predictive modeling. |

findings are as follows, which underscore both the promise and the necessity of adaptive sensing methodologies:

- **Accuracy gain:** Adaptive sensing significantly improves accuracy up to 47.58%p without model modification or maintain accuracy despite $50\times$ model size difference.

- **Synergy:** Adaptive sensing synergistically integrates with model improvement techniques.

- **Specificity:** Adaptive sensing must be tailored in a model- and scene-specific manner.

- **Human vs. Model optics:** High-quality images for model perception differ from those optimized for human perception.

Beyond image classification, adaptive sensing has shown consistent benefits across *diverse vision tasks*. In more complex settings such as object detection, segmentation, and remote photoplethysmography (rPPG), adaptive control of sensor parameters or viewpoints improves model robustness and accuracy under environmental shifts compared to non-adaptive configurations (e.g., fixed or auto-exposure, static viewpoints) [8, 89, 54]. In 6D object-pose estimation—a multimodal task involving both depth and RGB channels—adaptive cross-modal coordination marks a new stage of adaptive sensing by jointly optimizing multiple sensing modalities [28]. It shows that the multi-modal control can achieve higher accuracy and stability than single-modal controls (e.g., RGB-only or depth-only) and surpasses factory defaults, while further enhancing robustness and data efficiency. Together, these findings establish adaptive sensing as a scalable, complementary, and generalizable paradigm for robust perception across diverse sensing modalities and task conditions.

## 3.2 Advantages for Real-World Agents and Deployment

Integrating adaptive sensing yields practical advantages in realistic operational settings:

- **Learning perspective:** Unlike simulation environments [50, 24, 80], real-world environments pose challenges due to sensor heterogeneity (cameras, microphones, haptic arrays) and unpredictable conditions (lighting, weather) [44, 15, 35, 32]. Adaptive sensing enables agents to dynamically optimize sensor parameters (e.g., exposure, viewpoint), effectively reducing perceptual uncertainty. This targeted data acquisition approach enhances sample-efficient learning and robust generalization, especially under sparse reward conditions.

- **Engineering and Economic Viability:** Adaptive sensing reduces computational requirements compared to extensive model retraining, well-suited for resource-constrained and embedded systems. The improved data quality at the sensor level reduces infrastructure costs, enabling economically sustainable AI deployment at scale [85].

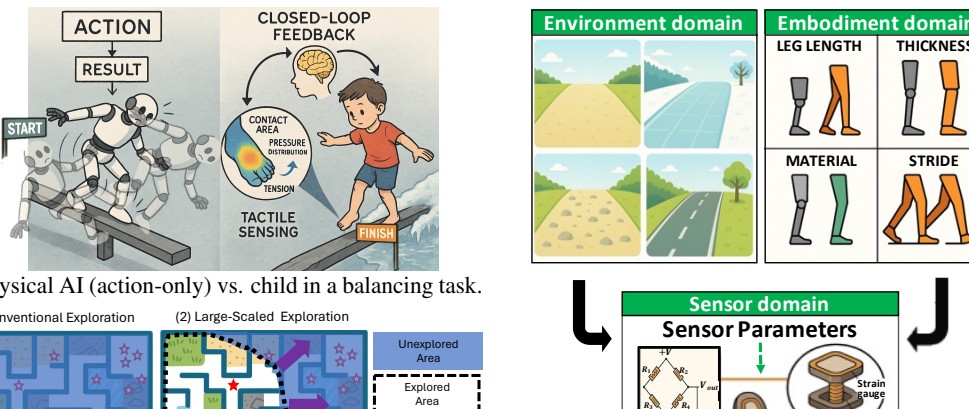

(a) Physical AI (action-only) vs. child in a balancing task.

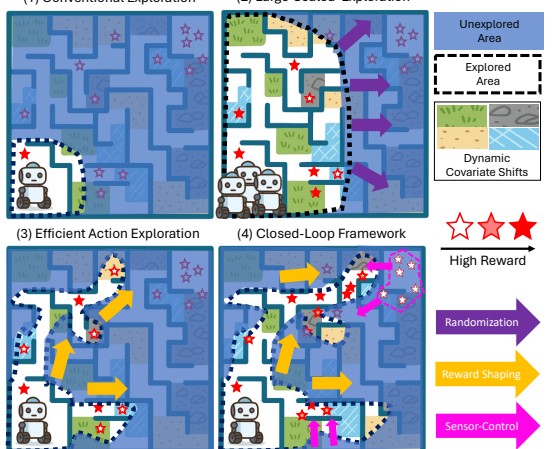

(b) Exploration in dynamic, sparse-reward settings. (1–3): Motor-only learning vs. (4): Perception-aware learning.

(c) Illustration of covariate shift in embodied AI settings for a balancing task.

Figure 2: Why Closed-Loop Adaptive Sensing Framework is needed for Embodied AI Agents?

- **Ethical and Societal Impact:** By enabling targeted, context-aware data collection, adaptive sensing mitigates biases prevalent in large, static datasets [68, 53, 22]. This ensures fairer outcomes in sensitive applications, enhancing public trust and ethical integrity in AI systems.

- **Interdisciplinary Innovation.** Adaptive sensing naturally encourages collaboration among computer scientists, engineers, ethicists, neuroscientists, and policymakers, fostering holistic solutions and comprehensive technological advancements.

- **Broad Domain Applicability:** Adaptive sensing has the potential to practically impact diverse domains that utilize sensor data. Concrete scenarios are provided in Table 2.

In summary, adaptive sensing represents a necessary paradigm shift for smaller, greener, and fairer AI systems—turning "sense better" into "learn less."

## 4    Adaptive Sensing for Embodied AI: Obstacles and Outlook

This section explores the deeper potential of adaptive sensing within embodied AI, expanding beyond the initial evaluation in static image classification tasks. Current embodied AI systems predominantly rely on *action-centric training* without explicitly considering adaptive sensing, resulting in inefficient learning and limited adaptability [20]. This limitation becomes particularly evident when comparing action-only embodied agents to humans, who seamlessly integrate sensory adaptation with motor actions in a closed-loop manner. For instance, consider a balancing task (Fig. 2a), which involves simultaneous covariate shifts [6, 90, 9] stemming from variations in environmental conditions, sensor interfaces and parameters, and agent morphologies (Fig. 2c), and extremely sparse reward signals relative to the complexity of the multi-sensor, multi-modal exploration space (Fig. 2b). Humans quickly achieve robust performance through efficient closed-loop sensory feedback, requiring only a few training trials and effortlessly adapting to new conditions. In contrast, embodied agents relying solely on motor actions significantly lag in both learning speed and real-world robustness.

**Current Gaps in Adaptive Sensing Approaches:** Existing adaptive sensing methods, notably Lens [8], predominantly target single-shot perception tasks (Fig. 3b), neglecting scenarios involving

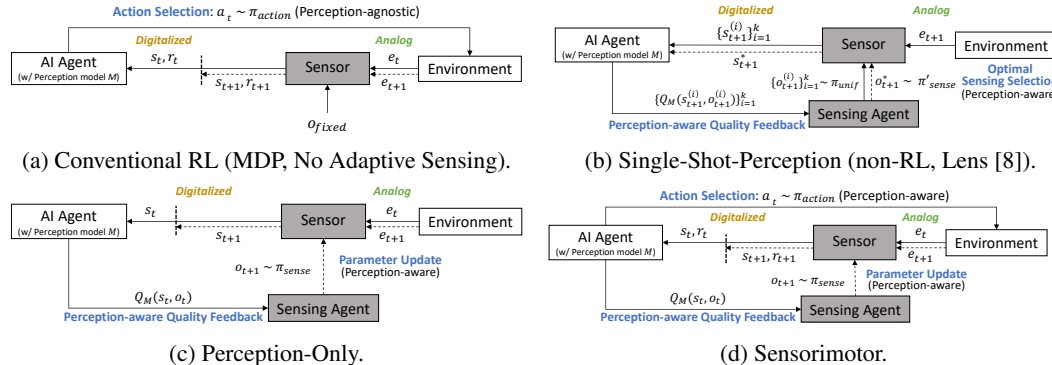

(a) Conventional RL (MDP, No Adaptive Sensing).

(b) Single-Shot-Perception (non-RL, Lens [8]).

(c) Perception-Only.

(d) Sensorimotor.

Figure 3: Towards Closed-Loop Adaptive Sensing Framework for Embodied AI Agents.

continuous interaction and sequential decision-making. These approaches fail to consider the crucial role of ongoing closed-loop feedback between sensing, perception, and action, which is essential for embodied agents operating within dynamic, real-world conditions. Moreover, current methods do not actively utilize model perception feedback to guide sensor parameter exploration. This limitation is especially critical in continuous, sparse-reward scenarios, where effective navigation of the sensor configuration space through closed-loop adaptive sensing is critical for robust and efficient learning. Addressing these gaps is imperative for successfully integrating adaptive sensing with continuous, action-driven learning paradigms in embodied AI.

## 5 Closed-Loop Framework for Embodied AI Agents: Towards Humanoid

To fill the gaps and integrate adaptive sensing into embodied AI across both reinforcement learning (RL) and non-RL settings, we propose a principled **Adaptive Sensing Framework**. The framework is *scalable* and covers *essential settings* for embodied AI, starting from the most fundamental RL setting, a standard MDP formulation [62] as sequential decision-making. It also encompasses non-RL settings, drawing on single-shot control scenarios explored in early adaptive sensing studies, and progressively extends to continuous perception tasks, sensorimotor interactions, and ultimately multimodal embodied scenarios, all within a fully closed-loop adaptive structure. The ultimate goal is to establish a robust, adaptive AI pipeline capable of efficiently learning and reliably adapting to dynamic environments.

### 5.1 RL Agents with Sensing–Environment Interaction

To pave the way for closed-loop adaptive sensing in embodied AI, we first formalize two baseline setups—each reflecting that the agent's state is observed by sensor measurements (via measurement function $f$) from the interaction between sensor configuration and the environment:

#### 5.1.1 Common Notations

**Basic Environment Components.** Given state space $\mathcal{S}$, action space $\mathcal{A}$, transition probability $P$, and reward function $R$, the environment response distribution $P_E(e_{t+1} \mid s_t, a_t)$ defines the probability of the next environmental state $e_{t+1}$ conditioned on the current state-action pair $(s_t, a_t)$, encompassing all potential covariate shifts arising within the environment domain as a consequence of the agent's actions. Similarly, the reward is generally sampled as: $r_{t+1} \sim R(s_t, \cdot)$ and optionally may incorporate additional contextual information such as sensing or action history.

**Sensor Parameter Options Space** $\mathcal{O}$ ($\subseteq \mathbb{R}^p$)**.** The set of all possible sensing settings for $p$ parameters. An option $o_t \in \mathcal{O}$ represents settings of sensing parameters at time $t$. The option $o_{\text{fixed}} \in \mathcal{O}$ denotes a constant parameter configuration, typically used in non-adaptive (fixed) sensing settings.

**Sensor Measurement** $f(e_t, o_t)$**.** A sensor operation converting analog signals from the environment $e_t$ into a digital observation $s_t$ under sensor configuration $o_t$ at timestep $t$. This process inherently captures various covariate shifts resulting from sensor-environment interactions.

**Perception-Aware Quality Estimation** $Q_M(s_t, \cdot)$**.** A perception-based metric evaluating how easily an observation $s_t$ can be interpreted by a given perception model (or module) $M$. For example, if $M$ is an image classification model, the quality metric can be defined as the maximum confidence score (Lens [8]): $Q_M(s_t) := \max(\text{softmax}(M(s_t)))$. Optionally, additional contextual information, such as sensing or action history, can be incorporated into this metric.

**Action Policy** $\pi_{\textbf{action}}(a_t \mid s_t, \cdot)$**.** General policy selecting the agent's action based on relevant states and contextual information such as action history (e.g., $\pi_{\text{action}}(a_t \mid s_t, a_{t-1})$).

**Sensing Policy** $\pi_{\textbf{sense}}(o_{t+1} \mid s_t, \cdot)$**.** General policy selecting the next sensing configuration conditioned on prior observations and contextual information such as sensing history, and perception-aware quality metrics (e.g., $\pi_{\text{sense}}(o_{t+1} \mid s_t, o_t, Q_M)$).

**Conventional RL (MDP) without Adaptive Sensing. (Fig. 3a)** We represent the environment as a standard Markov Decision Process (MDP) [62], $\mathcal{M} = (\mathcal{S}, \mathcal{A}, P_E, R)$ assuming non-adaptive (i.e., fixed) sensor configurations. At each timestep $t$, the agent executes:

1. **State Observation:**   $s_t \in \mathcal{S}$.

2. **Action Selection:**   $a_t \sim \pi_{action}(a_t \mid s_t)$.

3. **Environment Response and Sensor Measurement (No Sensing Agent):**

$$e_{t+1} \sim P_E(e_{t+1} \mid s_t, a_t), \quad s_{t+1} \sim f(e_{t+1}, o_{fixed}) \tag{1}$$

4. **Reward Collection:**   $r_{t+1} \sim R(s_t, a_t)$

**Single-Shot Adaptive Sensing (Perception-Only, No Actions, (Fig. 3b)).** We define a newly augmented stochastic process $\mathcal{P} := (\mathcal{S}, \mathcal{O}, P'_E, Q_M)$ defined by $P'_E$ which is the environment transition probability similar to $P_E$ defined in Eq.(1), but agnostic to actions. This new stochastic process $\mathcal{P}$ captures ways in which sensing configurations are adaptively selected per-timestep for single-shot perception tasks (e.g., image classification). Concretely, at timestep $t$, the agent performs:

1. **State Candidate Observation:** For a given candidate sensor configuration $o_{t+1}$ at timestep $t + 1$ and the current state $s_t$, a candidate sensor measurement is sampled via the following approach:

$$e_{t+1} \sim P'_E(e_{t+1} \mid s_t), \quad s_{t+1} \sim f(e_{t+1}, o_{t+1}). \tag{2}$$

2. **Perception-Aware Quality Estimation:** Evaluate the quality of the candidate measurement using the perception-aware metric, $Q_M(s_{t+1}, o_{t+1})$.

3. **Candidate Sampling (Perception-Agnostic Exploration):** To choose a well-performing sensor configuration $o^*_{t+1}$ for observing a promising state $s^*_{t+1}$, we first sample $k$ candidate configurations, where $k$ is a hyperparameter controlling the overall processing time. These configurations form a set: $O_{t+1} := \{o^{(1)}_{t+1}, o^{(2)}_{t+1}, \ldots, o^{(k)}_{t+1}\}$, sampled uniformly without replacement from a state-agnostic policy $\pi_{\text{unif}}$ over the sensor configuration space $\mathcal{O}$. For each sampled candidate $o^{(i)}_{t+1}$ ($i = 1, \ldots, k$), we obtain the corresponding candidate state $s^{(i)}_{t+1}$ according to Eq. (2). The set of candidate states is then defined as: $S_{t+1} := \{s^{(1)}_{t+1}, s^{(2)}_{t+1}, \ldots, s^{(k)}_{t+1}\}$.

4. **Optimal Sensing Selection (Perception-Aware Sensing Policy) and State Observation:** Finally, the perception-aware sensing policy $\pi'_{\text{sense}}$ selects the optimal sensing configuration $o^*_{t+1}$ among candidates based on the perception-aware quality metric $Q_M$: $o^*_{t+1} = \arg\max_{i \in \{1, \ldots, k\}} Q_M(s^{(i)}_{t+1}, o^{(i)}_{t+1}) =: \pi'_{\text{sense}}(O_{t+1} \mid S_{t+1}, Q_M)$.

   The agent then observes the perceptually optimal state using the chosen sensing configuration: $s^*_{t+1} \sim f(e_{t+1}, o^*_{t+1})$.

## 5.2   Single-Modal Continuous Perception Tasks

As shown in Fig. 3c, we generalize the single-shot adaptive sensing scenario (Lens; Fig. 3b) to a sequential, continuous reinforcement learning (RL) setting without explicit physical actions. We formalize this scenario as a Markov Decision Process (MDP), defined by $\mathcal{M} = (\mathcal{S}, \mathcal{O}, P'_E, Q_M)$, where the agent's decisions exclusively involve adaptively selecting sensing configurations $o_t \in \mathcal{O}$. At each timestep $t$, the agent performs the following:

1. **State Observation:**   Observe current state $s_t \in \mathcal{S}$.

2. **Perception-Based Sensing Selection:**   Evaluate perception-aware quality metric $Q_M(s_t, o_t)$ and select the next sensor configuration via policy $\pi_{\text{sense}}$: $o_{t+1} \sim \pi_{\text{sense}}(o_{t+1} \mid s_t, o_t, Q_M)$.

3. **Environment Response and Sensor Measurement:**

$$e_{t+1} \sim P'_E(e_{t+1} \mid s_t), \quad s_{t+1} \sim f(e_{t+1}, o_{t+1}).$$

In contrast to Lens, where sensing parameters were selected randomly or via simple heuristics, the sensing policy here explicitly maximizes the model-centric quality metric $Q$. Drawing inspiration from how infants progressively refine gaze control based on continuous perceptual feedback [92, 51], our closed-loop adaptive sensing strategy promotes efficient, coherent sensor exploration, resulting in robust and improved performance in single-modal continuous perception tasks.

## 5.3 Continuous Sensorimotor Tasks

**Key Intuition: Closed-loop Feedback Learning in Humans.** In real-world sensorimotor tasks, agents must dynamically co-adapt sensing parameters and physical actions to maintain balance and locomotion across varied terrains. Humans instinctively—or using tools—modulate sensor gains (e.g., foot-pressure thresholds) and adjust actions (e.g., stride lengths) according to environmental conditions such as icy, rocky, or uphill surfaces. Through repeated sensorimotor experiences within this closed-loop feedback framework, perception continuously informs actions, and actions shape subsequent sensory inputs, enabling robust, adaptive behavior.

We formalize this scenario as a Markov Decision Process (MDP) defined by $\mathcal{M} = (\mathcal{S}, \mathcal{A}, \mathcal{O}, P_E, Q_M)$, where the agent jointly selects sensing configurations $o_t \in \mathcal{O}$ and physical actions $a_t \in \mathcal{A}$. At each timestep $t$, the agent executes:

1. **State Observation:**  Observe current state $s_t \in \mathcal{S}$.
2. **Perception-Aware Action Selection:**  Select the next action via policy $\pi_{\text{action}}$, conditioned on perception-aware quality feedback: $a_t \sim \pi_{\text{action}}(a_t \mid s_t, o_t, a_{t-1}, Q_M)$.
3. **Perception-Aware Sensing Selection:**  Select the next sensor configuration via policy $\pi_{\text{sense}}$:
$$o_{t+1} \sim \pi_{\text{sense}}(o_{t+1} \mid s_t, o_t, Q_M).$$
4. **Environment Response and Sensor Measurement:**
$$e_{t+1} \sim P_E(e_{t+1} \mid s_t, a_t), \quad s_{t+1} \sim f(e_{t+1}, o_{t+1}).$$
5. **Reward Collection:**  Given a hyperparameter $\lambda$, the agent balances task-oriented performance with sensing-quality feedback, enabling efficient joint learning of sensing and action policies: $r_{t+1} \sim R(s_t, a_t, o_t) = R_{\text{task}}(s_t, a_t) + \lambda Q_M(s_t, o_t)$. where $R_{task}$ is a task-specific reward function that quantifies the effectiveness or success of the agent's actions in achieving the intended physical objective (e.g., duration of maintained balance).

**Adaptation to Multi-modality (Sensors) Settings.** Beyond adapting sensor parameters within a single modality, agents may also dynamically allocate attention across multiple sensory modalities depending on the task context. For example, when standing still, weight shifts toward the forefoot increase reliance on toe pressure sensors, whereas a lateral push engages ankle proprioceptors more heavily to restore balance. Formally, at each timestep $t$, we introduce a modality-weight vector $w_t \in \mathbb{R}^N$ across $N$ available sensory modalities (e.g., pressure, torque, IMU), typically normalized so that $\sum_{n=1}^N w_t[n] = 1$. Extending the single-modality sensing policy $\pi_{\text{sense}}$ and measurement function $f$, we define the multi-modality sensing policy $\pi_{\text{multi-sense}}$ and measurement function $f_{\text{multi-sense}}$ as:

$$(o_{t+1}, w_{t+1}) = \pi_{\text{multi-sense}}(o_{t+1}, w_{t+1} \mid s_t, o_t, w_t, Q_M), \quad s_{t+1} = f_{\text{multi-sense}}(s_t, o_t, w_t).$$

By adapting $w_{t+1}$ in response to environmental perturbations—such as increasing reliance on ankle sensors when experiencing lateral instability—the agent effectively focuses its sensing resources on the most informative modalities. This mechanism mirrors human sensorimotor reflexes, promoting rapid, robust adaptation and recovery.

**Humanoid-Scale Multimodal Tasks under Sparse Rewards.**  For challenging humanoid tasks—such as opening a bottle cap—agents typically receive a binary, sparse reward $R_{\text{sparse}} \in \{0, 1\}$ only upon successful task completion. To facilitate efficient exploration and learning under these sparse reward conditions, we introduce intermediate perception-aware, continuous sensing-quality metrics $Q_i$. These metrics quantify intermediate progress or cross-modal sensor feedback, thus providing informative and dense guidance.

For instance, if visual alignment is uncertain while closing a bottle cap, tactile feedback indicating increased grip tightness can enhance the visual sensing-quality metric. Conversely, uncertain tactile sensing can be clarified by visual information regarding precise object positioning. Formally, such cross-modal or intermediate-quality metrics are defined as follows:

$$Q_{\text{grip}}(s_t, a_{t-1}, o_t^{\text{tact}}), \quad Q_{\text{vis}}(s_t, a_{t-1}, o_t^{\text{cam}}, o_t^{\text{tact}}),$$

where $Q_{\text{grip}}$ measures grip stability from tactile sensors, and $Q_{\text{vis}}$ evaluates visual alignment accuracy informed by both visual and tactile sensor configurations.

These quality metrics can be integrated into a composite reward function:

$$R_t = R_{\text{sparse}} + \lambda_{\text{tact}} Q_{\text{grip}}(s_t, a_{t-1}, o_t^{\text{tact}}) + \lambda_{\text{vis}} Q_{\text{vis}}(s_t, a_{t-1}, o_t^{\text{cam}}, o_t^{\text{tact}}),$$

where hyperparameters $\lambda_{\text{tact}}$ and $\lambda_{\text{vis}}$ control the relative contribution of each modality. Each metric $Q_i$ explicitly captures task-relevant aspects of sensor data quality, such as grip stability, visual accuracy, uncertainty reduction, or information gain. By leveraging these intermediate, modality-specific metrics, the agent receives rich, continuous feedback tailored to downstream objectives, effectively mitigating the exploration challenges posed by sparse environmental rewards.

# 6 Challenges and Counterarguments

While adaptive sensing offers transformative potential to reshape AI design and operation, its widespread adoption faces several critical challenges and counterarguments.

## 6.1 Challenges

- **Lack of Benchmarks:** Existing benchmarks [19, 45, 56, 23] rarely capture sensor and environmental variations. Recent adaptive-sensing benchmarks [9, 8] remain narrowly focused on image classification tasks, limiting generalizability and broader methodological insights.

- **Unspecified Objectives:** Existing adaptive sensing strategies rely on metrics such as model confidence or out-of-distribution (OOD) scores [8], which inadequately capture data-quality under realistic covariate shifts. This hampers accurate assessments and practical applicability.

- **Complexity of Multi-modal Sensor Spaces:** Multi-modal sensors enlarge the perceptual and solution spaces, increasing learning complexity—especially under sparse or delayed reward conditions.

- **Performance Trade-offs:** Adaptive sensing strategies may at first underperform large-scale, resource-intensive models and may face real-time or deployment bottlenecks in practice. Stakeholders may prioritize immediate maximal accuracy over long-term sustainability and efficiency.

- **Integration Barriers with Existing Frameworks:** Implementing adaptive sensing may increase complexity and require significant modifications to existing AI pipelines, software stacks, and hardware platforms [1, 58, 88, 84], posing logistical, economic, and organizational challenges.

- **Ethical and Privacy Concerns:** Real-time, context-aware sensor optimization introduces potential ethical risks related to privacy, particularly when operating in sensitive environments.

## 6.2 Counterarguments

- **Proprietary Sensors & Transparency:** Although many sensors are proprietary, adaptive sensing does not require internal hardware access; existing APIs for exposure or gain control are sufficient. As demand grows, vendors are likely to expand such interfaces without compromising intellectual property. Overall, adaptive sensing is more likely to *increase*, rather than reduce, transparency.

- **Alternative methods:** Approaches such as efficient architectures and federated learning (FL), often proposed within the current model-centric paradigm to address generalization and computational challenges, cannot recover information lost at capture time or resolve environment–sensor shifts. They also introduce trade-offs—efficiency may reduce OOD robustness and fairness, while FL suffers from non-IID bias and privacy risks. Adaptive sensing complements these methods by operating upstream, allowing compact models to match or even surpass large-scale ones.

- **Task-dependent or rapidly changing configurations:** In single-task deployments, Lens selects from a small runtime candidate pool, so task dependence is not limiting. For multi-tasking, condition the pool on the active task with a conditional/hierarchical policy. Under dynamics, avoid per-frame updates by triggering switches on drift/performance proxies (e.g., confidence, calibration, quality drops), stabilize with hysteresis and switch-cost/latency budgets, and schedule changes via history-/resource-aware bandits or reinforcement learning.

- **When and Why Adaptive Sensing Excels:** Adaptive sensing outperforms domain adaptation, domain generalization, and test-time adaptation under *covariate shifts*, as hardware-level control directly reshapes raw measurements and can preserves information that post-hoc model adaptation cannot recover (§1). In contrast, under semantic or content shifts (e.g., unseen classes), model-level adaptation remains more effective. **The two paradigms are complementary**, addressing

orthogonal failure modes and achieving the best performance when combined. In *resource-constrained* settings, adaptive sensing remains attractive for its efficiency, enabling small models to rival much larger ones (§3). Determining when to apply each approach remains an open question (§7), as current OOD and uncertainty detectors primarily flag semantic shifts, leaving reliable differentiation between covariate and semantic (or mixed) shifts underexplored.

- **Why Closed-Loop? (vs. Separated):** A standalone controller (as in Lens [8]) is feasible and effective in low-dynamic environments where no action policy is involved (Fig. 3b). However, in embodied AI settings characterized by dynamic covariate shifts and sparse rewards, an optimal sensing strategy must be carefully designed to provide richer feedback and mitigate reward sparsity, as it depends on both the model's perception characteristics and task objectives. Closed-loop co-optimization aligns sensor control with the agent's policy learning, stabilizing observations and improving both sample efficiency and robustness (§4–5).

## 7 Open Research Directions

Addressing the identified challenges requires concerted investigation across the following strategic research avenues:

- **Standardized Benchmarks:** Develop comprehensive, realistic benchmarks that evaluate dynamic sensor-level adaptation for various tasks, environments, and modalities, and develop synthetic simulations informed by the real-world data, enabling fair, robust, and comparable assessments.

- **Data-quality Metrics:** Develop test-time evaluation metrics reflecting the model's perception of sensor data quality under covariate shifts, leveraging both aleatoric and epistemic uncertainty. This will enable effective guidance and informed decisions in adaptive sensing.

- **Algorithmic Innovation in Real-time Adaptation:** Advance efficient algorithms using reinforcement learning [77], adaptive control [41], and online learning [31] to optimize sensor parameters in dynamic environments. Developing frameworks to explore and exploit sensor configuration spaces will enhance responsiveness and efficiency. Future directions also include addressing practical deployment challenges such as multi-sensor synchronization, single-sensor multitasking, and deadline-constrained adaptation, which are key to mitigating real-time bottlenecks.

- **Co-Development of Models and Sensor Strategies:** Develop tightly integrated approaches that simultaneously optimize AI model architectures and adaptive sensor parameters, leveraging mutual feedback for improved generalization and robustness.

- **Multimodal and Language-Driven Adaptive Sensing:** Integrate language-based and multimodal context into sensor adaptation strategies, facilitating intuitive human-AI interaction and broader applicability across domains such as robotics, healthcare, and interactive systems.

- **Privacy-Preserving Sensor Optimization:** Investigate secure, lightweight, and privacy-aware adaptive sensing methods suitable for on-device deployment. Interdisciplinary collaboration involving AI experts, ethicists, and policymakers will be critical for maintaining public trust.

## 8 Conclusion

In this paper, we have advocated for adaptive sensing as a critical paradigm shift for overcoming fundamental limitations to the prevailing model-centric approach in AI. Drawing inspiration from robust biological sensory systems, adaptive sensing provides a practical pathway toward environmentally sustainable, computationally efficient, ethically responsible, and equitably distributed AI capabilities. By actively optimizing sensor parameters at the input stage, adaptive sensing significantly alleviates the computational and economic burdens currently in large-scale model training and deployment.

We have identified critical gaps in existing methodologies, their limited consideration of continuous, closed-loop interactions essential for real-world embodied agents. Addressing these gaps through targeted research—such as standardized adaptive benchmarks, real-time adaptation algorithms, privacy and ethical standards, and multimodal sensor integration—will be vital. There are urgent, interdisciplinary opportunities that the broader NeurIPS community is uniquely positioned to tackle.

Given the escalating environmental, economic, and societal pressures resulting from conventional AI scaling, embracing adaptive sensing is not simply advantageous; it is imperative. We call upon the NeurIPS community to prioritize this research agenda. Now is the moment for decisive collective action—by integrating adaptive sensing, we can ensure AI's trajectory aligns fundamentally with ecological responsibility, ethical integrity, and global equity.

## Acknowledgments

This work was supported in part by Institute of Information communications Technology Planning Evaluation (IITP) grant funded by the Korea government (MSIT) (No. RS-2025-02263754, RS-2025-25463302), in part by the National Research Foundation (NRF) of Korea grant funded by the Korea government (MSIT) (No. RS-2023-00222663).

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
