# OpenReview forum: "Position: AI Should Sense Better, Not Just Scale Bigger: Adaptive Sensing as a Paradigm Shift"
_NeurIPS.cc/2025/Position_Paper_Track — NeurIPS 2025 Position Paper Track_

### Official Review · Reviewer_L59A · 2025-07-23

**Significance:** 4
**Presentation:** 3
**Rating:** 7
**Confidence:** 4

**Summary:**

This paper champions the idea of "adaptive sensing" as a groundbreaking shift in how we develop AI, steering away from the usual focus on just scaling up neural models and datasets. Taking cues from how biological sensory systems work, the authors suggest that AI should focus on dynamically fine-tuning sensor parameters - like exposure, sensitivity, and multimodal setups - right at the input stage, instead of just relying on bigger models. They provide compelling evidence that adaptive sensing can help smaller models outperform much larger ones while also cutting down on computational costs. The paper introduces a closed-loop framework for embodied AI agents and lays out research paths for weaving adaptive sensing into various fields such as robotics, healthcare, and autonomous systems. The key takeaway is that this approach paves the way for a more sustainable, efficient, and fair AI landscape.

**Strengths:**

The paper presents an engaging vision for sustainable AI through adaptive sensing, supported by some promising initial evidence. The biological inspiration is not only well-justified but also clearly articulated. It offers a thorough roadmap that covers various application areas, complete with tangible examples. The writing style is approachable, making it easy for readers without deep technical knowledge to grasp the concepts. This topic tackles pressing issues regarding the environmental and societal effects of AI. Additionally, the proposed framework effectively connects different research fields. The formal mathematical foundation for embodied AI provides a solid technical basis.

**Weaknesses:**

The evidence we have mainly comes from image classification tasks, and it shows some limitations in generalization. This paper is quite lengthy for a position paper, which might make it less accessible to some readers. Some of the claims regarding the potential of adaptive sensing could be a bit exaggerated, especially considering the limited evidence available. The section on challenges could do a better job of addressing the fundamental limitations we face. Additionally, alternative methods for sustainable AI, like efficient architectures and federated learning, don’t get much attention. It seems like the complexity of putting adaptive sensing into practice in real-world systems might be underestimated. A more detailed analysis of when adaptive sensing is truly beneficial would really bolster the argument.

**Questions:**

How does the computational overhead of continuously adapting sensors stack up against the claimed efficiency gains, especially in environments with limited resources?

What specific benchmarks or evaluation frameworks could effectively showcase the advantages of adaptive sensing across various tasks, beyond just image classification?

How would adaptive sensing hold up in situations where the best sensor configurations are highly dependent on the task at hand or change quickly?

**Alternative Position:**

Yes, and alternative positions are well-considered and named but not addressed

**Author Identification:**

No.

**Context:**

4

**Discussion:**

4

**Ethics:**

["NO or VERY MINOR ethics concerns only"]

**Position:**

Yes, the paper argues for or against a position related to machine learning.

**Support:**

3

**Thoroughness:**

4

---

### Official Review · Reviewer_RAQz · 2025-08-04

**Significance:** 3
**Presentation:** 2
**Rating:** 5
**Confidence:** 3

**Summary:**

This paper argues for adaptive sensing as a fundamental paradigm shift away from the model-centric adaptation and generalization in the current mainstream AI communities. The authors draw inspiration from biological sensory systems that dynamically adjust parameters, such as pupil size and focus. The central claim in the paper is that current foundation models' reliance on scaling compute and datasets creates unsustainable environmental, economic, and ethical costs while failing to address real-world domain shifts efficiently and effectively.
The authors mainly support their position through evidence from existing computer vision domains, theoretical arguments about biological sensing superiority, and extensive impactful applications spanning robotics, healthcare, and autonomous systems. The paper calls for systematic research integration of adaptive sensing into embodied AI systems, proposing multimodal realtime frameworks, identifying challenges, and outlining research directions for the community.

**Strengths:**

1. The biological inspiration is well-motivated and provides an intuitive framework for understanding why adaptive sensing could be more efficient in some aspects.
2. The authors have effectively outlined the limitations and challenges, covering environmental sustainability, economic accessibility, generalization failures, and ethical concerns.
3. The potential integration of adaptive sensing in embodied AI contexts is very intriguing.

**Weaknesses:**

1. The paper primarily relies on computer vision classification task as the main empirical results to support the claims. While the Lens framework shows promising results, this represents evaluation on a narrow set of controlled scenarios. The lack of comprehensive evaluation across tasks, sensor/modality configurations, and real-world deployment conditions weakens the argument  for broad adoption.
2. The paper doesn't adequately address when and why adaptive sensing would outperform adaptive modeling approaches. Instead of controlling sensors, adaptation layers for diverse sensing inputs or models that generalize across domains might be more practical solutions.
3. The authors do not sufficiently address the practical challenges of implementing adaptive sensing in existing AI pipelines. The hardware requirements, real-time computational overhead, and software infrastructure needs are mentioned but not thoroughly analyzed. This gap makes it difficult to assess the feasibility of the proposed paradigm shift beyond research prototypes.
4. Although Section 6 identifies key challenges, the response to the potential counterargument is relatively shallow.

**Questions:**

1. How could we systematically distinguish between cases where poor performance stems from sensor limitations versus model inadequacies?
2. Why does the framework require co-development of models and sensor strategies rather than treating adaptive sensor control as a separate, broadly applicable module? Could adaptive sensor parameter optimization be developed as a standalone component that works across different downstream models, similar to how data augmentation or preprocessing pipelines operate independently?
3. Given that domain adaptation, robust training methods, and model ensembling have shown significant success. When could adaptive sensing provides advantages over these alternatives?
4. Can the authors discuss relevant real-time and deployment bottlenecks?
5. The authors need to clarify when sensor limitations have been exceeded versus when model adaptation alone would suffice. A systematic analysis comparing sensor-level versus model-level adaptation would better justify the proposed approach and clarify when each strategy is optimal.

**Alternative Position:**

Yes, and alternative positions are trivial straw-man arguments

**Author Identification:**

No.

**Context:**

2

**Discussion:**

3

**Ethics:**

["NO or VERY MINOR ethics concerns only"]

**Position:**

Yes, the paper argues for or against a position related to machine learning.

**Support:**

2

**Thoroughness:**

4

---

### Official Review · Reviewer_co5F · 2025-08-08

**Significance:** 3
**Presentation:** 3
**Rating:** 4
**Confidence:** 4

**Summary:**

The position paper argues for more adaptive approaches as a paradigm shift in AI/ML. In comparison model-based approaches that make predictions based on a static set of learned weights, adaptive sensing methods can optimize parameters (e.g., camera exposure or positioning) based on the input. A number of advantages of such methods are described in Section 3.2, and include more sample efficient learning (i.e., less training examples) and reduced uncertainty.

**Strengths:**

The position is states and the advantages and disadvantages of the position are described. The manuscript is generally easy to follow and understand.

**Weaknesses:**

The mathematics of reinforcement learning (RL) introduced in Section 5 is complex. In general, while the paradigm of adaptive sensing proposed in the introduction is very convincing, RL is just one of the ways how it can be accomplished. It wasn’t clear to me why the RL based approach is described in such detail then. Other approaches that the authors may consider include:

Raisi (2019). Physics-informed neural networks: A deep learning framework for solving forward and inverse problems involving nonlinear partial differential equations.

Maier (2018). Deep scatter estimation (DSE): feasibility of using a deep convolutional neural network for real-time x-ray scatter prediction in cone-beam CT.

These and other methods combine both physics-based modeling and AI to achieve the best of both worlds, using AI to accelerate slow computational processes in modeling, and can also be seen as adaptive sensing methods. In summary, I believe the paper would be much stronger if either the position was revised to only focus on RL or other, non-RL approach discussion would be incorporated into the manuscript.

**Questions:**

It is not clear how the proposed paradigm shift will take into account the proprietary nature of most sensors, i.e., it will be difficult to design such algorithms  when majority of manufacturers do not share details of their sensors. Moreover, the proposed shift may encourage any adaptive sensing AI algorithms to become proprietary (no model weights shared) themselves, negatively affecting AI transparency and safety efforts.

An existing method (Lens [6]) is described. Are there other examples of adaptive sensing techniques that exist that are similar?

It is not clear if “adaptive” would refer to digital replicas of sensors that are rooted in physics/optics, or learning-based techniques that replicate the behavior seen in physical sensors.

**Alternative Position:**

Yes, and alternative positions are well-considered and named but not addressed

**Author Identification:**

No.

**Context:**

2

**Discussion:**

3

**Ethics:**

["NO or VERY MINOR ethics concerns only"]

**Position:**

Yes, the paper argues for or against a position related to machine learning.

**Support:**

4

**Thoroughness:**

4

---

### Note · Authors · 2025-09-05

**1-10 Additional Comments:**

I felt it was a bit unfortunate that there was no discussion period, as such an exchange could be especially valuable for a new track like the position paper track, where arguments are shaped around fresh perspectives. At the same time, I understand this was the first year of the track at NeurIPS, and there were likely many challenges in organizing it. Submitting to this track and receiving reviews was a valuable opportunity to view our work from different angles and refine our ideas more carefully. I would like to sincerely thank the chairs for making this venue possible.

**1-11 Submit Again:**

Unsure

**1-1 Submission Process:**

3

**1-2 Next Year:**

I hope there is enough time for authors and reviewers to have in-depth discussions.

**1-3 Future Development:**

Please consider requiring that each submitted paper contributes at least one co-author to serve as a reviewer for the track, ensuring a sufficient and fair reviewer pool.

**1-4 Interest:**

["Other (please specify in the next question)"]

**1-4 Other Interest:**

Not at this time.

**1-5 Thoughtful:**

6

**1-6 Supportive:**

7

**1-7 Technical Aspects Versus Position:**

7

**1-8 Gate Keeping:**

7

**1-9 Camera Ready Changes:**

We thank all of the reviewers for positive reviews, and for highlighting our strengths:
- Writing & outline (Unanimous): clear, well-structured, approachable.
- Position (RAQz, L59A): well-motivated biological inspiration tied to societal challenges.
- Framework  (RAQz, L59A): intuitive, solid, bridges to diverse fields.
- Potential (RAQz, L59A): intriguing vision with a thorough roadmap toward sustainable AI.
- Evidence  (RAQz, L59A): promising results showing efficiency and strong performance.

We sincerely thank the reviewers for constructive comments. We addressed all concerns with additional analysis and clarifications, summarized below:
- Evidence & Benchmarks beyond classification (All): Clarified adaptive sensing is not limited to classification, with prior work showing benefits in segmentation, detection, and multimodal tasks.
- Potential Counterarguments (All): Addressed concerns on (i) proprietary sensors & transparency (co5F): adaptive sensing works via software APIs and may enhance transparency; (ii) real-time & deployment bottlenecks (RAQz, L59A): outlined as open research directions; (iii) alternative methods (L59A): noted they are model-centric and complementary, not replacements; (iv) task-dependent/dynamic configs (L59A): specified when valid and proposed concrete mitigations and future directions.
- Adaptive Sensing Strategy (RAQz): Explained when/why it outperforms—strong under covariate shifts, complementary to model-level methods under semantic shifts, and appealing in resource-constrained settings—while noting open research directions; also clarified the rationale for a closed-loop design.
- Scope Misunderstanding (co5F): Reiterated that adaptive sensing is real hardware-level control, applicable in both RL and non-RL contexts, guided by model needs rather than human perception; the concern reflects scope misalignment, not inherent limitations.

We will revise the final version to highlight key strengths and incorporate these clarifications.

**3-1 Review Response1:**

co5F

**3-2 Reaction To Review1:**

We would like to clarify several points, including a key misunderstanding. As outlined below, we will revise the manuscript to clarify these distinctions and present our contributions with greater precision and transparency.

[W1/Q3: Clarification of Our Position and Framework]
In our work, adaptive sensing refers to physical, sensor-level control before digitization (e.g., exposure, viewpoint), not physics-based simulation or model-level methods (e.g., PINNs, DSE). As §1 explains, how raw signals are digitized is critical: information lost at capture cannot be recovered, while simulation/post-hoc fixes—tuned for human perception not for DNNs—remain brittle under environment–sensor shifts. In contrast, direct hardware control optimizes measurements for the model and improves robustness. Our framework is scalable and not RL-only: §5 highlights RL as a mainstream paradigm for sequential decision making in embodied AI that has yet to incorporate adaptive sensing, while also covering non-RL single-shot control (e.g., Lens; Fig. 3b). In short, adaptive sensing is hardware-level control, applicable in both RL and non-RL contexts and guided by model needs, not human perception. These concerns reflect scope misalignment rather than inherent limitations.

[Q1: Proprietary sensors & transparency]
While many sensors are proprietary, adaptive sensing does not require access to hardware internals. Existing software APIs (e.g., exposure, gain) suffice for research and deployment, and rising demand will push vendors to extend them without risking IP. Sensor makers are not model developers; their value is enabling open software ecosystems with the AI community. Thus this paradigm is more likely to increase—rather than reduce—transparency and safety.

[Q2: Beyond Lens]
Lens is one instance among several, with prior work in tasks such as detection (Baek et al., ICLR 2025; Wong et al., NSDI 2024) and multimodal (Han et al., arXiv) settings, supporting the paradigm’s generality.

**3-3 Review Response2:**

RAQz

**3-4 Reaction To Review2:**

Thank you for the feedback. We will clarify these points in revision. The concerns do not undermine our position and will be addressed with evidence and deployment discussion.

[W1: Evidence beyond classification]
Adaptive sensing is not limited to classification; prior work shows gains in segmentation, detection (Baek et al., ICLR 2025; Wong et al., NSDI 2024), and multimodal tasks (Han et al., arXiv). We will add concise results and citations.

[Q4 / W3 / W4: Real-time & Deployment Bottlenecks]
We will expand §6–7 to analyze these bottlenecks and outline concrete directions (multi-sensor synchronization,  single-sensor multi-tasking, deadline constraints), while noting open challenges.

[W2 / Q1 /  Q3 / Q5: When/Why adaptive sensing outperforms]
Adaptive sensing outperforms TTA/DG under covariate shift, as hardware-level control reshapes raw measurements (e.g., correcting exposure) and preserves information that post-hoc model adaptation cannot recover. Under semantic/content shifts (e.g., unseen classes), model-level adaptation (DA/DG/TTA) is preferable. The two are complementary, addressing orthogonal failure modes and performing best together. In resource-constrained settings, adaptive sensing is especially appealing: lightweight parameter control with negligible overhead yields large gains, letting small models surpass ones up to 50× larger (Lens). Choosing which lever to pull remains open (§7), as current OOD/uncertainty detectors mainly flag semantic shifts, and reliable triage between covariate vs. semantic (or mixed) shifts is underexplored.

[Q2: Why closed-loop? (vs. separated)]
A standalone controller is feasible and already covered as the separated case (Lens-style), which works well in low-dynamics settings (Fig. 3b). In embodied AI with dynamic covariate shift and sparse rewards, sensing is model/task-dependent; closed-loop co-development aligns sensing with the policy, stabilizes observations, and improves sample efficiency and robustness (§4,5).

**3-5 Review Response3:**

L59A

**3-6 Reaction To Review3:**

Thank you for the thoughtful feedback. We will clarify and strengthen the manuscript by tightening §2 (counterarguments; position vs. efficient architectures and FL), refining §6–7 (deployment bottlenecks with concrete directions), cataloging open issues in the Supplementary, and adding compact evidence and benchmarks beyond classification, while clarifying task- and dynamics-aware control.

[W1,Q2: Evidence & Benchmarks beyond classification] Adaptive sensing extends beyond classification; prior work reports gains in segmentation, detection (Baek et al., ICLR 2025; Wong et al., NSDI 2024), and multimodal tasks (Han et al., arXiv). We will add concise quantitative summaries with citations and integrate the corresponding benchmarks/protocols to substantiate this.

[W2, W3: Alternative Methods] Efficient architectures and federated learning (FL) are useful but remain model-centric; they cannot recover capture-time information loss or environment–sensor covariate shifts—the core issues we target—and, as discussed in §2, they introduce trade-offs (efficiency may degrade OOD robustness and group fairness; FL shifts energy/communication to the edge, suffers non-IID/participation bias, and raises privacy/poisoning risks). Adaptive sensing operates upstream, complements these methods, and—when paired with compact models—can match or exceed much larger models (see §1).

[Q1, Q3: Task-dependent or rapidly changing configurations]
In single-task deployments, Lens selects from a small runtime candidate pool, so task dependence is not limiting. For multi-tasking, condition the pool on the active task with a conditional/hierarchical policy. Under dynamics, avoid per-frame updates by triggering switches on drift/performance proxies (e.g., confidence, calibration, quality drops), stabilize with hysteresis and switch-cost/latency budgets, and schedule changes via history-/resource-aware bandits or RL.

---

### Meta-Review · Area_Chair_JLXB · 2025-09-18

**Rating:** 7
**Confidence:** 4

**Strengths:**

Reviewers agree that the biological inspiration is a good motivation for the work and that the paper in general is well written. Reviewers also feel that the topic is important and relevant for the NeurIPS community to consider, as it cogently pushes back against a currently dominant and successful paradigm.  The paper successfully takes a persuasive position on an under-considered issue.

**Weaknesses:**

Reviewers agree that the tight focus on computer vision is a limitation of the current iteration of the paper; it is not always clear how the claims made would generalize to other single or multi-modal models.
The authors address part of this by committing to adding evidence for tasks beyond classification, including segmentation and detection, but it would be helpful to spend more time on how well this would generalize to non-vision models. For example, reviewer RAQz asks for consideration of additional "sensor/modality configurations."

**Questions:**

Precursors to this idea have been considered in the history of AI research, especially in the early days of adaptive sensing and adaptive optics (1950s - 70s). Might the authors consider adding a mention of the historical context of this idea in the opening of the paper or in Section 6 as a way to explain what has changed to allow the paradigm to succeed now?

**Ethics:**

None raised

**Thoroughness:**

4

---

### Decision · Program_Chairs · 2025-09-26

Accept